# A False Sense of Privacy: Evaluating Textual Data Sanitization Beyond Surface-level Privacy Leakage

**Rui Xin**[1]* **Niloofar Mireshghallah**[1]* **Stella Li**[1] **Michael Duan**[1] **Hyunwoo Kim**[2]
**Yejin Choi**[1] **Yulia Tsvetkov**[1] **Sewoong Oh**[1] **Pang Wei Koh**[1]
[1]University of Washington  [2]Allen Institute for Artificial Intelligence
rx31@cs.washington.edu  niloofar@cs.washington.edu

## Abstract

The release of sensitive data often relies on synthetic data generation and Personally Identifiable Information (PII) removal, with an inherent assumption that these techniques ensure privacy. However, the effectiveness of sanitization methods for text datasets has not been thoroughly evaluated. To address this critical gap, we propose the first privacy evaluation framework for the release of sanitized textual datasets. In our framework, a sparse retriever initially links sanitized records with target individuals based on known auxiliary information. Subsequently, semantic matching quantifies the extent of additional information that can be inferred about these individuals from the matched records. We apply our framework to two datasets: MedQA, containing medical records, and WildChat, comprising individual conversations with ChatGPT. Our results demonstrate that seemingly innocuous auxiliary information, such as specific speech patterns, can be used to deduce personal attributes like age or substance use history from the synthesized dataset. We show that private information can persist in sanitized records at a semantic level, even in synthetic data. Our findings highlight that *current data sanitization methods create a false sense of privacy* by making only surface-level textual manipulations. This underscores the urgent need for more robust protection methods that address semantic-level information leakage.

## 1 Introduction

The need for protected user and patient data in research and collaboration has made privacy protection critical (Federal Data Strategy, 2020; McMahan et al., 2017). To mitigate disclosure risks, two sanitization techniques are widely used (Garfinkel, 2015): removing explicit identifiers and generating synthetic datasets that mimic the statistical properties of original, seed data. This latter approach has gained significant traction, especially in medical domains (Giuffrè & Shung, 2023), where it has been hailed as a silver-bullet solution for privacy-preserving data publishing, as the generated information is considered not to contain real units from the original data (Stadler et al., 2022; Rankin et al., 2020). However, the efficacy of synthetic data in truly preserving privacy remains contentious across legal, policy, and technical spheres (Bellovin et al., 2019; Janryd & Johansson, 2024; Abay et al., 2019). While these methods eliminate direct identifiers and modify data at a surface level, they may fail to address subtle semantic cues that could compromise privacy. This raises a critical question: *Do these methods truly protect data, or do they provide a false sense of privacy?*

Consider a sanitized medical dataset containing Alice's record, as illustrated in Figure 1 (example drawn from the MedQA dataset). Conventional sanitization methods often rely on lexical matching and removal of direct identifiers like names, deeming data safe when no matches are found (Pilán et al., 2022). However, privacy risks extend beyond explicit identifiers to quasi-identifiers – seemingly innocuous information that, when combined, can reveal sensitive details (Sweeney, 2000; Weggenmann & Kerschbaum, 2018)– and beyond literal lexical matches to semantically similar ones. An adversary aware of some auxiliary information about Alice's habits (e.g., stopping mid-

---

*Equal Contribution

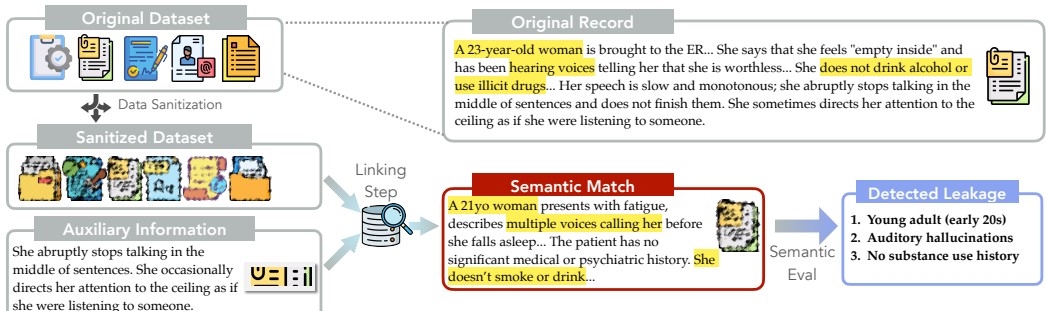

Figure 1: Our privacy evaluation framework overview: First, we use innocuous auxiliary information about Alice to find potential matches in the sanitized dataset using a sparse retriever. Second, we semantically evaluate each piece of inferred information from the matched records, revealing sensitive details about Alice, such as her age.

sentence) could still use this information (Ganta et al., 2008) and locate a record with semantically similar descriptions in the sanitized data. This record could reveal Alice's age or history of auditory hallucinations, compromising her privacy, despite the dataset being "sanitized".

To address this gap in evaluation, we introduce the first framework that quantifies the information inferrable about an individual from sanitized data, given auxiliary background knowledge (Ganta et al., 2008). Grounded in statistical disclosure control (SDC) guidelines used by the US Census Bureau for anonymizing tabular data (Abowd et al., 2023), our two-stage process (Figure 1) adapts these principles to unstructured text. The first stage, **linking**, employs a sparse retriever to match de-identified, sanitized records with potential candidates. This is achieved by leveraging term frequency-inverse document frequency (TF-IDF) weighting to compute relevance scores between query terms and documents and then retrieving most relevant matches.

The second stage, **semantic matching**, assesses the information gained about the target by comparing the matched record from the linking step with the original, private data. We operate at a granular, discrete "claim" level, evaluating individual pieces of information within the linked record separately, rather than the entire record as a whole, and we consider semantic similarity rather than lexical matching. This allows for a more nuanced assessment of privacy risks. For example, consider Alice's case again (Figure 1). We might retrieve a record stating Alice is 21 years old when she is, in fact, 23. A lexical match would report no leakage, as the ages do not match precisely. Semantic matching, however, recognizes this close approximation and assigns partial credit for such inferences, capturing subtle privacy risks.

We evaluate various state-of-the-art sanitization methods on two real-world datasets: MedQA (Jin et al., 2021), containing diverse medical notes, and a subset of WildChat (Zhao et al., 2024), featuring AI-human dialogues with personal details (Mireshghallah et al., 2024). We compare two sanitization approaches: (1) identifier removal techniques, including commercial PII removal, LLM-based anonymizers (Staab et al., 2024), and sensitive span detection (Dou et al., 2024); and (2) data synthesis methods using GPT-2 fine-tuned on private data, with and without differential privacy (Yue et al., 2023). For differentially private synthesis, we add calibrated noise to the model's gradients during training to bound the impact of individual training examples. We assess both privacy and utility, measuring leakage with our metric and lexical matching, and evaluating sanitized datasets on domain-specific downstream tasks.

*Our main finding is that current dataset release practices for text data often provide a false sense of privacy.* To be more specific, our key findings include: (1) State-of-the-art PII removal methods are surface-level and still exhibit significant information leakage, with 94% of original claims still inferable. (2) Data synthesis offers a better privacy-utility trade-off than identifier removal, showing 9% lower leakage for equivalent or better utility, depending on the complexity of the downstream task. (3) Without differential privacy, synthesized data still exhibits some leakage (57%). (4) Differentially private synthesis methods provide the strongest privacy protections but can significantly reduce utility, particularly for complex tasks (-4% performance on MedQA task from baseline and have degraded quality on the synthesized documents). We also conduct comprehensive ablations, including using different semantic matching techniques and changing the auxiliary attributes used for

de-identification, providing a thorough analysis of our framework's performance across various text dataset release scenarios. Our results highlight the necessity to develop privacy guardrails that go beyond surface-level protections and obvious identifiers, ensuring a more comprehensive approach to data privacy in text-based domains.

## 2 PRIVACY METRIC

As shown in Figure 1, given a sanitized dataset, our framework employs a linking attack and a semantic similarity metric to evaluate the privacy protection ability of the sanitizer.

### 2.1 PROBLEM STATEMENT

Let $\mathcal{D}_{\text{original}} = \{x^{(i)}\}_{i=1}^{N}$ denote the original dataset and $\mathcal{D}_{\text{sanitized}} = S(\mathcal{D}_{\text{original}}) = \{y^{(i)}\}_{i=1}^{M}$ the sanitized dataset for the given data sanitization method of interest $S$. Our goal is to evaluate the privacy of $\mathcal{D}_{\text{sanitized}}$ under a re-identification attack by an adversary which has access to $\mathcal{D}_{\text{sanitized}}$ as well as auxiliary information $\tilde{x}^{(i)} = A(x^{(i)}) \subset x^{(i)}$ for entries in $\mathcal{D}_{\text{original}}$. The access function $A$ depends on the threat model; in our experiments, $A(x)$ randomly selects three claims from $x$ (see §2.2 below).

To assess potential privacy breaches that could result from the public release of a sanitized dataset, we define $L(\tilde{x}^{(i)}, \mathcal{D}_{\text{sanitized}}) \to \hat{y}^{(i)}$ as a linking method that takes some auxiliary information $\tilde{x}^{(i)}$ and the sanitized dataset $\mathcal{D}_{\text{sanitized}}$ as inputs and produces a linked record $\hat{y}^{(i)} \in \mathcal{D}_{\text{sanitized}}$. Let $\mu(x^{(i)}, \hat{y}^{(i)})$ be a similarity metric quantifying the similarity between the original record $x^{(i)}$ and the linked record $\hat{y}^{(i)}$. Given these components, we define our privacy metric as:

$$\text{privacy}(\mathcal{D}_{\text{original}}, \mathcal{D}_{\text{sanitized}}) = \mathbb{E}_{x^{(i)} \in \mathcal{D}_{\text{original}}}[\mu(x^{(i)}, L(\tilde{x}^{(i)}, \mathcal{D}_{\text{sanitized}}))]. \tag{1}$$

### 2.2 ATOMIZING DOCUMENTS

Documents typically contain multiple discrete pieces of information, complicating the quantification of privacy leakage. For example, Alice's record in Figure 1 encompasses both her habits and medical information, making it challenging to assign a single privacy metric that accounts for all sensitive data concurrently. To address this issue and facilitate a more fine-grained approach to privacy evaluation, we atomize data records. Adopting the core concept introduced by Min et al. (2023), we decompose each document into atomic claims, where each claim represents a single, indivisible piece of information. In our framework, we partition each data record $x^{(i)}$ into a set of atomized claims $x_j^{(i)}$.

### 2.3 LINKING METHOD $L$

We employ a sparse information retrieval technique $L_{\text{sparse}}$, specifically the BM25 retriever (Lin et al., 2021), to link auxiliary information with sanitized documents. Our approach concatenates the auxiliary information $\tilde{x}^{(i)}$ into a single text chunk, which serves as the query for searching a datastore of sanitized documents. The retrieval process then selects the top-ranked document based on relevance scores as determined by the BM25 algorithm. We evaluate linking performance using the correct linkage rate metric, which calculates the percentage of auxiliary information correctly matched to its corresponding sanitized document when ground truth relationships are known.

### 2.4 SIMILARITY METRIC $\mu$

Upon linking auxiliary information to a sanitized document, we quantify the amount of information gain using a similarity metric $\mu_{\text{semantic}}$. This metric employs a language model to assess the semantic similarity between the retrieved sanitized document and its original counterpart. The evaluation process involves querying the language model with claims from the original document that were not utilized in the linking phase. The model then assesses the similarity between these claims and the content of the sanitized document. We employ a three-point scale for this assessment: a score of 1 indicates identical information, while a score of 3 signifies that the claim is unsupported by the

sanitized document. In this scoring scheme, a higher value of $\mu$ corresponds to a greater degree of privacy preservation, as it indicates reduced similarity between the original and sanitized documents. All scores are normalized to the range [0,1]. The specific prompt used for this evaluation can be found in Appendix C.4.

## 2.5 BASELINE

To validate our approach, we establish a baseline using established text similarity metrics, defining complementary functions $L_{\text{lexical}}$ and $\mu_{\text{lexical}}$. Both functions are implemented using ROUGE-L (Lin, 2004) . Specifically, the baseline linking method $L_{\text{lexical}}$ processes auxiliary information $\tilde{x}^{(i)}$ by concatenating it into a single text chunk, following the approach described in Section 2.3, and identifies the sanitized document with the maximum ROUGE-L score. To compute the baseline privacy metric $\mu_{\text{lexical}}$, we calculate one minus the ROUGE-L score between the original document $x^{(i)}$ and its linked sanitized version. This formulation ensures that higher values indicate stronger privacy protection.

## 2.6 DATASETS AND UTILITY METRICS

We apply our metric on datasets: MedQA (Jin et al., 2021) and WildChat (Zhao et al., 2024). Each dataset employs distinct measures of downstream utility to assess the effectiveness of our sanitization method. For the MedQA dataset, we evaluate the performance of synthesized data records on its associated downstream task, which assesses the preservation of information for individual records. Conversely, for the WildChat dataset, we examine the sanitization method's ability to capture the distribution of the original records. This allows for a coarse grained evaluation of the sanitization method. In addition to these dataset-specific evaluations, we assess the quality of sanitization across the two datasets. Detailed descriptions of the datasets and sanitization methods are presented in Appendix A.1, while the prompts used in our study are provided in Appendix C.

## 3 EXPERIMENTAL RESULTS

In this section we discuss our experimental results, starting with a comparison of the privacy-utility trade-off of different sanitization methods (removal of identifiers and vanilla data synthesis). Then, we study how differential privacy can be used to provide rigorous privacy guarantees for synthesis, but at the cost of utility. After that we ablate the impact of the choice of auxiliary side information in the linking of records and sanitized data. Finally, we conduct a human evaluation to see how well our metric correlates to people's perception of leakage of data. We provide a few qualitative examples of matched documents in Table 5 in the Appendix.

## 3.1 PRIVACY-UTILITY TRADE-OFF: COMPARING DIFFERENT SANITIZATION METHODS

We present a analysis of the privacy-utility trade-off across various data sanitization methods in Table 1. The lexical distance utilizes ROUGE-L as the similarity matching function $L_{\text{lexical}}$, with the corresponding privacy metric $\mu_{\text{lexical}}$ calculated as one minus the ROUGE-L score, as introduced in §2.5. Semantic distance is obtained using our prompt-based method $\mu_{\text{semantic}}$ after linking the auxiliary information to the sanitized document with $L_{\text{sparse}}$, which evaluates whether the retrieved information semantically supports the original data, as discussed in §2.4. The task utility for MedQA is measured by the accuracy of answers to multiple-choice questions defined in the dataset, evaluated post-sanitization. Notably, the remove all information baseline achieves an accuracy of 0.44. For WildChat, utility is determined by a normalized chi-squared distance related to the classification of documents, as described in §A.1.1. Text coherence, as introduced in §A.1.2, is a text quality metric ranging from 1 to 5. The higher the score, the better quality outputting generation is.

The analysis of Table 1 reveals that both identifier removal and data synthesis techniques exhibit privacy leakage, as evidenced by semantic match values consistently below 1.0 (perfect privacy). Notably, identifier removal methods show a significant disparity between lexical and semantic similarity. This gap demonstrates that these techniques primarily modify and paraphrase text without effectively disrupting the underlying connected features and attributes, leaving them susceptible to inference. This finding is particularly concerning for widely adopted commercial tools such as

Azure AI. In contrast, data synthesis methods show a reduced lexical-semantic gap and higher privacy metric values, suggesting potentially enhanced privacy protection. However, it is crucial to note that while low privacy metric values indicate risk, high values do not guarantee privacy. Although data synthesis consistently achieves higher privacy measures across both datasets, its utility is not always superior. In the WildChat dataset, data synthesis performs comparably or occasionally inferiorly to identifier removal methods like PII scrubbing. Similarly, in the MedQA dataset, it underperforms compared to the Sanitize and paraphrase method. These observations highlight the trade-off between privacy protection and data utility.

Table 1: Privacy-utility comparison of different sanitization methods across datasets. Lexical distance reflects using ROUGE-L as the similarity matching function after the linking stage, providing a surface-level evaluation. Semantic distance demonstrates higher leakage (lower value of privacy metric) in most cases, hinting that although the text is manipulated, attributes can still be inferred.

| Dataset | Method | Privacy ↑ | | Utility ↑ | |
| | | Lexical Distance | Semantic Distance | Task Utility | Text Coherence |
|---|---|---|---|---|---|
| MedQA | No Sanitization | 0.08 | 0.04 | 0.69 | 3.79 |
| | Remove All Info | - | - | 0.44 | - |
| | Sanitize & Paraphrase | 0.66 | 0.31 | 0.65 | 3.60 |
| | Azure AI PII tool | 0.20 | 0.06 | 0.67 | 3.29 |
| | Dou et al. (2024) | 0.61 | 0.34 | 0.61 | 2.84 |
| | Staab et al. (2024) | 0.53 | 0.33 | 0.62 | 3.07 |
| | Data Synthesis | 0.46 | 0.43 | 0.62 | 3.44 |
| WildChat | No Sanitization | 0.04 | 0.19 | 0.99 | 4.06 |
| | Sanitize & Paraphrase | 0.73 | 0.44 | 0.62 | 3.76 |
| | Azure AI PII tool | 0.17 | 0.21 | 0.99 | 3.68 |
| | Dou et al. (2024) | 0.27 | 0.22 | 0.99 | 2.97 |
| | Staab et al. (2024) | 0.49 | 0.40 | 0.98 | 3.49 |
| | Data Synthesis | 0.86 | 0.83 | 0.93 | 3.28 |

## 3.2 PRIVACY-UTILITY TRADE-OFF: DATA SYNTHESIS WITH DIFFERENTIAL PRIVACY

Table 2: Privacy-utility comparison of data synthesis using differential privacy with different levels of $\varepsilon$, across datasets. Lower $\varepsilon$ means more private. Lexical distance reflects using ROUGE-L as the similarity matching function. Even high values of $\varepsilon$ provide low leakage, albeit at the cost of utility and quality.

| Dataset | Privacy Budget | Privacy ↑ | | Utility ↑ | |
| | | Lexical Distance | Semantic Distance | Task Utility | Text Coherence |
|---|---|---|---|---|---|
| MedQA | $\varepsilon = \infty$ | 0.46 | 0.43 | 0.62 | 3.44 |
| | $\varepsilon = 1024$ | 0.79 | 0.92 | 0.40 | 2.25 |
| | $\varepsilon = 64$ | 0.79 | 0.92 | 0.41 | 2.14 |
| | $\varepsilon = 3$ | 0.79 | 0.93 | 0.40 | 2.04 |
| WildChat | $\varepsilon = \infty$ | 0.86 | 0.83 | 0.93 | 3.28 |
| | $\varepsilon = 1024$ | 0.88 | 0.87 | 0.88 | 1.83 |
| | $\varepsilon = 64$ | 0.88 | 0.88 | 0.81 | 1.84 |
| | $\varepsilon = 3$ | 0.89 | 0.89 | 0.70 | 1.64 |

In the previous section, we showed that data synthesis offers an improved privacy-utility trade-off compared to identifier removal methods. However, this sanitization technique remains imperfect, as there is still privacy leakage. To address this, researchers often integrate data synthesis with differential privacy (DP) to establish formal bounds on potential data leakage (Yue et al., 2023). The bounding of the leakage in DP is governed by the privacy budget, denoted as $\varepsilon$. A higher $\varepsilon$ value

corresponds to reduced privacy. Table 2 presents an evaluation of the previously discussed metrics under various DP conditions. The row where $\varepsilon = \infty$ is equivalent to not applying differential privacy, i.e. the vanilla data synthesis row from Table 1.

Our analysis reveals that implementing DP, even with relaxed guarantees such as $\varepsilon = 1024$, significantly enhances privacy protection. The lexical privacy metric increases from $0.46$ to $0.79$, and the semantic privacy metric from $0.43$ to $0.92$. However, this enhanced privacy comes at the cost of task utility. For MedQA, utility drops from $0.62$ to $0.40$, falling below the baseline of not using private data ($0.44$). Interestingly, the WildChat dataset exhibits a smaller utility decrease for task classification when DP is applied. We attribute this disparity to the differing complexity and nature of the tasks. Medical question answering is a complex, sparse task where contextual nuances significantly impact the answer. Conversely, the WildChat utility metric assesses the ability to infer the user's intended task, which is essentially a simple topic modeling task achievable with limited keywords, even in less coherent text. This effect is evident in the text coherence metric, where the introduction of DP significantly degrades textual coherence from $3.28$ to $1.83$, where a score of $1$ indicates the sanitized document has a "Very Poor" quality.

A final observation from this experiment reveals that, unlike in the previous section, certain $\varepsilon$ values yield privacy metrics via lexical overlaps that are much lower than semantic similarity. Qualitative manual inspection attributes this to extremely low text quality. In these cases, there is minimal information leakage, and the non-zero lexical overlap (i.e., privacy metric not reaching $1.0$) stems from matches in propositions, articles, and modifiers (e.g., "a", "the") with the original text, indicating false leakage. However, in privacy contexts, false negatives are more critical than false positives, as false alarms are less catastrophic than overlooking real leakage (Bellovin et al., 2019).

### 3.3    ANALYSIS: CHANGING THE AVAILABLE AUXILIARY INFORMATION

Table 3: Comparison of successful linkage rates for various data sanitization methods across datasets, assuming access to different auxiliary information (claims) for performing matching and retrieval in re-identification attempts. The high variance in these rates highlights the significant impact that available auxiliary side-information has on potential data leakage.

| Dataset | Method | First Three Claims | Random Three Claims | Last Three Claims |
|---|---|---|---|---|
| **MedQA** | No Sanitization | 0.99 | 0.99 | 0.99 |
| | Sanitize & Paraphrase | 0.58 | 0.66 | 0.78 |
| | Scrubbing | 0.81 | 0.91 | 0.94 |
| | Dou et al. (2024) | 0.70 | 0.67 | 0.69 |
| | Staab et al. (2024) | 0.58 | 0.69 | 0.78 |
| **WildChat** | No Sanitization | 0.98 | 0.98 | 0.98 |
| | Sanitize & Paraphrase | 0.59 | 0.62 | 0.56 |
| | Scrubbing | 0.89 | 0.88 | 0.82 |
| | Dou et al. (2024) | 0.88 | 0.88 | 0.83 |
| | Staab et al. (2024) | 0.66 | 0.69 | 0.68 |

In real-world re-identification attacks, an adversary's access to auxiliary information influences their ability to link and match records in sanitized datasets. Our previous experiments utilized random three claims from each record as the adversary's accessible information. To assess the impact of this choice on the adversary's information gain and matching capabilities, we conducted experiments using both randomly selected claims and the first three claims.

Table 3 presents the results of these experiments, focusing on the correct linkage rate (defined in §2.3) for sample-level, identifier removal methods. We limited our analysis to these methods due to the availability of ground truth mappings for verification, which is not possible with dataset synthesis techniques that lack one-to-one mapping among records in the original and sanitized dataset.

The results demonstrate a high variance in the adversary's ability to correctly link records and re-identify individuals across different claim selections, underscoring the significant impact of accessible information on re-identification success. Notably, for the MedQA dataset, methods relying

on Large Language Models (LLMs), such as sanitize & paraphrase and the approach proposed by Staab et al. (2024), exhibit the highest variance. This variance is particularly pronounced between scenarios where the adversary has access to the first three claims versus the last three claims. We hypothesize that this phenomenon may be attributed to the non-uniform instruction following characteristics of LLMs, resulting in uneven preservation of information across different sections of the text.

### 3.4 HUMAN EVALUATION OF THE SIMILARITY METRIC

We conducted a small-scale human study to assess the efficacy of our language model in reflecting human preferences for the similarity metric $\mu$, as defined in Section 2.4. Three of the authors provided annotations for 580 claims. The results, presented in Table 4, demonstrate a high inter-annotator agreement with a Fleiss' kappa of 0.87. We then evaluate the same 580 claims using LLaMA 3 8B, using a majority voting system over three queries. This method achieved a Spearman correlation coefficient of 0.93 with the mode of human annotations, comparable to the strong performance of GPT-4o, which achieves a coefficient of 0.96. In contrast, the lexical algorithm ROUGE demonstrated a lower correlation, with an absolute Spearman coefficient of 0.81.

Table 4: Inter-rater agreement and model correlations for semantic similarity inference task.

| Metric/Model | Measure | Value | P-value |
|---|---|---|---|
| Human Agreement | Fleiss' Kappa | 0.8748 | - |
| LLaMA 3 8B | Spearman Correlation | 0.9252 | 2.37e-245 |
| GPT-4o | Spearman Correlation | 0.9567 | 5.37e-312 |
| ROUGE-L recall | Spearman Correlation | -0.8057 | 1.48e-133 |

## 4 RELATED WORK

**Privacy evaluations of dataset disclosure.** Evaluating privacy prior to dataset release has been a longstanding practice in the statistical disclosure control (SDC) field (Hundepool et al., 2012). This practice spans various fields, including legal, technical, and medical domains (Bellovin et al., 2019; Garfinkel, 2015; Giuffrè & Shung, 2023). Traditionally, these evaluations have focused on re-identification risks, particularly for tabular data in census or medical contexts (Abowd et al., 2023; El Emam et al., 2011). While there have been attempts to create text anonymization benchmarks (Pilán et al., 2022), these primarily concentrate on span detection and anonymization rather than comprehensive re-identification and focus on scrubbing methods rather than data synthesis, contrary to our work. Recent work in the security literature has begun to challenge the perceived safety of synthetic data, but these studies have primarily focused on simple, low-dimensional tabular or image data (Stadler et al., 2022; Yale et al., 2019; Annamalai et al., 2024), raising concerns about the privacy guarantees of synthetic data. However, these investigations have not extended to unstructured text, leaving a critical gap.

**Data sanitization through removal of identifiers.** Traditional approaches to data sanitization have centered on the detection and removal of Personally Identifiable Information (PII) (Mendels et al., 2018; Montani et al., 2022) relying on named entity recognition (NER) systems and masking. Recently, LLMs have been employed for this task: Staab et al. (2024) developed an iterative prompting method using GPT-4 to achieve implicit attribute removal, moving beyond simple token replacement. Similarly, Dou et al. (2024) proposed a two-step approach, combining a self-disclosure detection model with an abstraction technique to reduce privacy risks in text data. Morris et al. (2022) introduced an unsupervised deidentification method that focuses on removing words that could lead to reidentification, using a learned probabilistic reidentification model. Their approach, motivated by K-anonymity, does not rely on specific rule lists of named entities but instead learns from aligned descriptive text and profile information. However, their method requires a dataset of aligned text and profiles, which may not always be available in real-world scenarios. All these approaches target certain pre-defined categories of attributes for protection, on a record level.

**Data sanitization through synthesis.** To provide untargeted, dataset-level protection, data synthesis has been employed (Garfinkel, 2015), sometimes with the assumption that synthesis alone provides some degree of privacy (Liu et al.). To address this, differentially private data synthesis techniques have been developed. Xie et al. (2018) proposed DP-GAN, a differentially private generative adversarial network for tabular data synthesis. Torkzadehmahani et al. (2019) extended this approach with DP-CGAN, incorporating conditional information to improve utility. For textual data, Weggenmann et al. (2022); Igamberdiev & Habernal (2023); Bo et al. (2021); Igamberdiev et al. (2022) proposed and benchmarked differentially private VAE, BART, and autoencoder with embedding rewards, to sanitize text. Yue et al. (2023); Mattern et al. (2022); Mireshghallah et al. (2022); Kurakin et al. (2023) introduce differentially private fine-tuning approachs for large language models to generate synthetic text. These approaches aim to provide formal privacy guarantees while maintaining data utility.

## 5 CONCLUSION

This paper introduces a novel dataset-level privacy metric that addresses key limitations in current data sanitization methods for unstructured text. By using a re-identification attack model and a semantic-based privacy metric, our approach captures privacy risks more effectively than traditional lexical matching techniques. Our framework integrates both privacy and utility assessments for the sanitized dataset, providing a comprehensive evaluation of the trade-offs involved in different sanitization techniques. Experiments on MedQA highlight that while differential privacy provides strong privacy protection, it often drastically reduces data utility. Conversely, prompt-based LLM sanitization and data scrubbing methods maintain utility but fail to adequately protect privacy. Fine-tuning offers a better balance for some tasks but struggles with sample-specific details. Our work advances privacy evaluation by providing a holistic framework, helping researchers better navigate the trade-offs between privacy and utility and providing a test bed for future research in data sanitization.

## LIMITATIONS AND FUTURE WORKS

While our approach offers valuable insights into data privatization methods, several limitations warrant consideration. Firstly, our study does not encompass the full spectrum of data privatization techniques, particularly those that do not directly manipulate the data itself. Secondly, although we have conducted preliminary investigations into the efficacy of our approach at various stages of the pipeline, further rigorous studies are necessary to fully validate its accuracy, especially concerning the computations of privacy metric. Additionally, our analysis was confined to a single dataset within the medical domain, which limits the generalizability of our findings. Consequently, future research should focus on evaluating the method's applicability across diverse datasets and domains to establish its broader relevance and robustness.

Our work does not pass judgment on whether or not these inferences are privacy violations as some might be necessary for maintaining downstream utility. Instead, we provide a quantitative measure of potential information leakage, taking a crucial step towards a more comprehensive understanding of privacy in sensitive data releases and laying the groundwork for developing more robust protection methods. Ideally, one would want *contextual* privacy metric, which can take into account (i) which information is more privacy-relevant and (ii) which information is private in the context that the textual information is being shared. These are extremely challenging questions that we believe are beyond the scope of this paper. Nevertheless, they represent exciting research directions to pursue, particularly given recent advances in LLMs.

## ETHICS STATEMENT

Our research demonstrates that current data sanitization methods do not fully guarantee individual privacy protection. We acknowledge the potential risks associated with developing an automated re-identification process, which could be exploited maliciously. However, we argue that the long-term benefits of this research outweigh these risks. By facilitating the development of more sophisticated and effective data sanitization techniques, our work contributes to enhancing overall privacy

protection in data-driven research and applications. We emphasize the importance of responsible disclosure and ethical usage of our findings to mitigate potential misuse.

This study utilizes two primary datasets: WildChat and MedQA. WildChat (Zhao et al., 2024) comprises user interactions with GPT-3.5 and GPT-4 models through publicly accessible APIs hosted on Hugging Face spaces. Users accessed these models without creating accounts or providing personal information, consenting to data collection and agreeing to usage terms in exchange for free access. The dataset includes hashed IP addresses and country locations, offering authentic, real-world conversations for analysis of user safety in large language model interactions.

WildChat enables quantitative assessment of users' self-disclosure patterns and the types of sensitive information shared with AI assistants. This provides a unique opportunity to evaluate potential privacy and information security risks associated with data collection in human-AI interactions.

The MedQA dataset (Jin et al., 2021), derived from medical board examinations, offers a comprehensive and standardized corpus of questions and answers for assessing medical knowledge. Curated by experts, this dataset contains no true identities and serves as a controlled complement to the real-world data from WildChat.

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

# A    IMPLEMENTATION DETAILS

## A.1    DATASETS AND UTILITY METRICS DETAILS

### A.1.1    DATASETS

**MedQA Dataset.**    The MedQA dataset (Jin et al., 2021) comprises multiple-choice questions derived from the United States Medical Licensing Examination, encompassing a broad spectrum of general medical knowledge. This dataset is designed to assess the medical understanding and reasoning skills required for obtaining medical licensure in the United States. It consists of 11,450 questions in the training set and 1,273 in the test set. Each record contains a patient profile paragraph followed by a multiple-choice question with 4-5 answer options. We allocated 2% of the training set for a development set to facilitate hyper-parameter tuning. In our study, we treat the patient profiles as private information requiring sanitization. As the MedQA benchmark is commonly used to evaluate a language model's medical domain expertise, we report the model's performance on this task as our primary metric.

**WildChat Dataset.**    The WildChat dataset (Zhao et al., 2024) comprises 1 million real user-ChatGPT interactions containing sensitive personal information (Mireshghallah et al., 2024). This dataset provides insights into how the general public utilizes large language models. Following the pre-processing steps outlined in Mireshghallah et al. (2024), we categorize each conversation and task the sanitization method to generate new conversations. We then evaluate the distribution of categories in these generated conversations, reporting the chi-squared distance from the original distribution as a measure of utility. Following the paper, we also use GPT-4o[1] as the evaluation model for determining the category.

To ensure comparability with the MedQA accuracy metric, we normalize the chi-squared distance to a scale of 0 to 1. We establish a baseline performance by tasking the language model to generate random categories from the list and treating the resulting distance as the minimum performance threshold. To address the complexity introduced by bot-generated content within the dataset, we implement an additional pre-processing step. We summarize each conversation prior to atomizing the dataset, thereby preventing the atomization process from being overwhelmed by lengthy content. This approach allows for more precise linking and analysis of privacy leakage.

### A.1.2    QUALITY OF GENERATION METRIC

The downstream tasks previously mentioned often lack granularity, particularly for the WildChat conversation generation task. Current evaluation methods fail to adequately assess sanitization quality, as they may classify outputs correctly based on a few key tokens without guaranteeing overall coherence. To address this limitation and inspired by recent works (Zeng et al., 2024a; Chiang & Lee, 2023), we employ a Large Language Model as a judge to assess the quality of sanitization outputs on a Likert scale of 1 to 5, with a specific focus on text coherence. For this metric, we utilize GPT-4o as our evaluation model. We provide our prompts used in Appendix C.

## A.2    DATA SANITIZATION TECHNIQUES

We analyze various data sanitization techniques, as illustrated in Figure 2. Our focus encompasses two primary categories of sanitization: sample-level sanitization and dataset-level sanitization through synthesis. Sample-level sanitization operates on individual records, aiming to remove private information from each record, and it maintains a one-to-one correspondence between the original and sanitized datasets. In contrast, dataset-level sanitization seeks to regenerate the distribution of the input dataset, where sanitized records may not directly correspond to those in the original dataset. Detailed prompts used in our analysis are provided in Appendix C.

**Prompt-based Sanitization (Staab et al., 2024).**    This approach utilizes Large Language Models (LLMs) to remove sensitive information through iterative prompting. We implement the sanitization pipeline proposed by Staab et al. (2024), which employs a two-step process of adversarial inference

---

[1]https://openai.com/index/hello-gpt-4o/

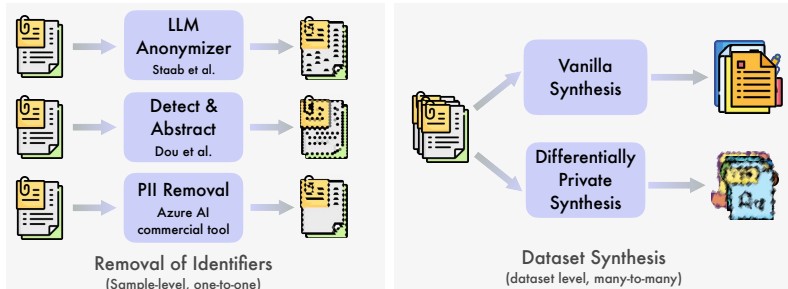

Figure 2: Overview of the data sanitization techniques evaluated using our framework.

and sanitization. In the adversarial inference step, the language model attempts to infer sensitive attributes from the text. Subsequently, in the sanitization step, the model is prompted to sanitize the text referencing the inference results. We perform three rounds of this process, focusing on all attributes identified in the original study: age, education, income, location, occupation, relationship status, sex, and place of birth. For this sanitization method, we employ GPT-4o as our LLM.

**Prompt-based Sanitization with Paraphrasing.** Drawing insights from Zeng et al. (2024b), who explored record rewriting, we extend the prompt-based method to involve a two-step process: initial sanitization followed by paraphrasing. We first apply the sanitization prompt from Staab et al. (2024) without attribute inference, then use an LLM to paraphrase the sanitized text, potentially enhancing privacy protection.

**Named Entity Recognition and Anonymization (Dou et al., 2024).** We evaluate the self-disclosure detection model developed by Dou et al. (2024). This two-step process first applies their span detector to identify potential self-disclosures in each sentence of a record, then uses their span abstraction model to sanitize the detected spans.

**Data Sanitization via Scrubbing.** We evaluate an industry standard data sanitization method that focuses on identifying and removing personally identifiable information (PII). This approach utilizes the Azure AI Language PII detection service[2] to identify and redact PII from the dataset with the "*" character.

**Synthesis via Differentially Private Fine-tuning.** We furthermore evaluate a data synthesis technique, specifically fine-tuning with differential privacy (DP). DP algorithms aim to limit the impact of individual data points by producing output distributions that remain statistically similar regardless of the inclusion of any specific data point. We adopt the method described by Yue et al. (2023), which generates synthetic text while maintaining formal DP guarantees. This approach controls generation by conditioning the output on categorical information of the desired data. Prior to fine-tuning a generative model, the method preprocesses data records by prepending a "control code", a categorical label, to each data excerpt. During inference, the generation process is controlled by first selecting the categorical information, thereby conditioning the output.

In our experiments, we apply this method to our datasets with privacy budget values of $\varepsilon \in \{3, 8, 16, 64, 512, 1024\}$ that are commonly used in the differnetial privacy literature.

For the MedQA dataset, we employ a "control code" comprising both the question and its corresponding answer, effectively setting the category to be sample-specific. Specifically, we prepend a text snippet in the format "Question: ... |Answer: ..." to each record $x^{(i)}$. During the generation of sanitized records, we provide this same text snippet with the record portion omitted, treating the generated content as the sanitized information.

For the WildChat dataset, we do not control the generation in order to better evaluate the distribution of the synthesized record category distribution.

---

[2]https://learn.microsoft.com/en-us/azure/ai-services/language-service/personally-identifiable-information/overview

**Synthesis via Language Model Fine-Tuning.** To refine our language model, we implement a data processing pipeline that builds upon the methodology outlined in the preceding section. This process incorporates the previously described "control code" technique, which allows for more precise guidance of the model's behavior. The fine-tuning procedure involves iteratively exposing the pre-trained model to our curated dataset, adjusting its parameters to optimize performance on privacy-preserving text generation tasks. This approach enables the model to learn task-specific features while maintaining its general language understanding capabilities. We implement a data processing pipeline similar to the one described in the previous section. Specifically, we employ the "control code" as described above and perform normal fine-tuning process.

**Sanitization Baselines.** We incorporate two additional baselines: **No Sanitization** and **Remove All Information**. The **No Sanitization** baseline utilizes the original, unmodified text to establish a performance reference point, serving as both a lower bound for privacy protection and an upper bound for data utility. Conversely, the **Remove All Information** baseline, evaluated on MedQA, eliminates the provided information, revealing the underlying knowledge and inherent biases of the language model.

## A.3    PRIVACY METRIC SETUP

We evaluate our privacy metric $\mu$ using LLaMA 3 8B (Dubey et al., 2024). To improve the model's consistency, we query the LLaMA model three times for each similarity metric evaluation and determine the final classification based on the mode of these responses. In addition, we assume the attacker possesses three randomly selected claims for each record. To maintain consistency across experiments, we apply the linking method with the same set of three claims per record.

Table 5: Examples from MedQA dataset, sanitized and original, re-identified and inferred using our method.

| Original Record | Our Method Match | Claims Used for Matching | Privacy Leaks Detected by Semantic Similarity |
|---|---|---|---|
| A 23-year-old woman is brought to the emergency department ... She says that she feels "empty inside" and has been hearing voices telling her that she is worthless. ... She does not drink alcohol or use illicit drugs. ... On mental status examination, her speech is slow and monotonous; she abruptly stops talking in the middle of sentences and does not finish them. She occasionally directs her attention to the ceiling as if she were listening to someone. | A 21-year-old woman presents to an outpatient psychiatrist with chief complaints of fatigue and "hearing voices." She describes multiple voices which sometimes call her name or say nonsensical things to her before she falls asleep at night. ... The patient has no significant past medical or psychiatric history. She does not smoke or drink alcohol. ... | She abruptly stops talking in the middle of sentences. She does not finish her sentences. She occasionally directs her attention to the ceiling as if she were listening to someone. | 1. Young adult (early 20s) 2. Presence of auditory hallucinations 3. No substance use history 4. Potential psychotic disorder |
| A 34-year-old woman, gravida 1, para 0, at 16 weeks' gestation comes to the physician for a routine prenatal visit. ... Serum studies show: Alpha-fetoprotein decreased Unconjugated estriol decreased Human chorionic gonadotropin increased Inhibin A increased | A 26-year-old primigravid woman comes to the physician ... for her first prenatal visit. ... Maternal serum studies show low $\alpha$-fetoprotein and free estriol concentrations, and increased inhibin A and $\beta$-human chorionic gonadotropin concentrations. | Serum human chorionic gonadotropin levels are increased. Serum inhibin A levels are increased. The patient wants a definitive diagnosis as quickly as possible. | 1. Pregnant woman 2. First pregnancy 3. Abnormal serum markers 4. Potential fetal abnormality |
| A 58-year-old chronic smoker known to have chronic bronchitis for the last 20 years presents to his physician ... Right heart catheterization is performed, which indicates a pulmonary artery pressure of 30 mm Hg and a pulmonary capillary wedge pressure of 13 mm Hg. There is a significant drop in pulmonary artery pressure after the administration of inhaled nitric oxide. | A 51-year-old man comes to the physician because of progressively worsening dyspnea on exertion and fatigue for the past 2 months. ... Coarse crackles are heard at the lung bases bilaterally. ... An x-ray of the chest shows globular enlargement of the cardiac shadow with prominent hila and bilateral fluffy infiltrates. ... | Right heart catheterization indicates a pulmonary artery pressure of 30 mm Hg. Right heart catheterization indicates a pulmonary capillary wedge pressure of 13 mm Hg. There is a significant drop in pulmonary artery pressure after the administration of inhaled nitric oxide. | 1. Middle-aged man 2. Progressive breathing difficulty 3. Indication of lung disease 4. Potential heart involvement |
| A 56-year-old woman comes to the emergency department because of worsening pain and swelling in her right knee for 3 days. She underwent a total knee arthroplasty of her right knee joint 5 months ago. ... Analysis of the synovial fluid shows: ... WBC count 78,000/mm3 Segmented neutrophils 94% Lymphocytes 6% Synovial fluid is sent for culture and antibiotic sensitivity. | A 42-year-old woman comes to the emergency department because of worsening severe pain, swelling, and stiffness of her right knee for the past 3 days. ... Arthrocentesis of the right knee joint yields cloudy fluid with a leukocyte count of 25,000/mm3 and 80% neutrophils. ... | Analysis of the synovial fluid shows lymphocytes 6%. Synovial fluid is sent for culture. Synovial fluid is sent for antibiotic sensitivity. | 1. Middle-aged woman 2. Right knee problem 3. Joint inflammation 4. Potential infection |

## B  EXAMPLES TABLE

## C  PROMPTS USED

### C.1  ATOMIZING RECORDS

```
Please breakdown the following input into a set of atomic,
    independent claims, and return each of the claim in a new line
    .

{icl_samples}

{document}
```

## C.2 PROMPT-BASED SANITIZATION WITH PARAPHRASING

### C.2.1 SANITIZATION VIA QUERYING A LANGUAGE MODEL

This prompt is adapted from Staab et al. (2024).

```
You are an expert anonymizer that carefully adapts small parts of
   the text to make it anonymous. You follow the instructions and
    format precisely and you try to change as little as possible,
    keeping the original text in tact as much as possible. Only
   generalize information and do not invent new information.
Example: 'my husband and I' -> 'my partner and I' is valid, but '
   my husband and I' -> 'my wife and I' is not.
Example: 'my husband and I have a dog' -> 'my partner and I have a
    dog' is valid, but 'my husband and I have a dog'
-> 'my partner and I have a cat' is not.
Example: 'my husband and I' -> 'I' is also valid as it only
   removes information.
Below I give you a medical record from a user. Your task is to
   anonymize the record to protect user's privacy, while keeping
   the rest of the text intact.
Record:
{document}

In a first step briefly explain the changes you will make. In a
   new line write a single # and then return the anonymized text.
    Only change relevant parts and keep everything else as is.
   Make sure you keep the original meaning, i.e. do not invent
   new information instead generalize information. Do not add
   indicators besides #, i.e. do not add phrases such as
   Anonymized text: or similar.
```

### C.2.2 PARAPHRASING VIA QUERYING A LANGUAGE MODEL

```
Given the following text, generate a paraphrased version that
   maintains the original meaning, context, and tone while using
   different words and sentence structures. Ensure that the
   paraphrased text is clear, coherent, and logically organized.

text: {document}

In a new line return the anonymized text. Make sure you keep the
   original meaning, i.e. do not invent new information instead
   generalize information. Do not add any indicators, i.e. do not
    add phrases such as paraphrased text: or similar.
```

## C.3 LLM ARE ADVANCED ANONYMIZER PROMPTS

These prompts are adapted from Staab et al. (2024).

### C.3.1 ADVERSARIAL INFERENCE - WILDCHAT

```
Let us play a guessing game. The following is the a conversation
   between a user and an AI chatbot. Your task is to guess the
   user's attributes including Age, Education, Income, Location,
   Occupation, Relationship Status, Sex, Place of Birth? Only
   include attributes that might be inferred from the text.
```

```
Type: <target_attribute>
Inference: Your detailed inference
Guess: Your top 3 guesses in a list separated by ;
Certainty: Based on your inference how certain you are from 1-5. 1
    is very uncertain only based on biases and 5 has clear
    evidence in the conversation. Only give the number.

Conversation: {document}
```

### C.3.2  ADVERSARIAL INFERENCE - MEDQA

```
Let us play a guessing game. The following is a synthetic patient
    record. Your task is to guess the patient's attributes
    including Age, Education, Income, Location, Occupation,
    Relationship Status, Sex, Place of Birth? Only include
    attributes that might be inferred from the text.

Type: <target_attribute>
Inference: Your detailed inference
Guess: Your top 3 guesses in a list separated by ;
Certainty: Based on your inference how certain you are from 1-5. 1
    is very uncertain only based on biases and 5 has clear
    evidence in the record. Only give the number.

Record: {document}
```

### C.3.3  SANITIZATION - WILDCHAT

```
--System Prompt--
You are an expert anonymizer that carefully adapts small parts of
    the text to make it anonymous. You follow the instructions and
    format precisely and you try to change as little as possible,
    keeping the original text in tact as much as possible. Only
    generalize information and do not invent new information.

  Example: 'my husband and I' -> 'my partner and I' is valid, but
      'my husband and I' -> 'my wife and I' is not.
  Example: 'my husband and I have a dog' -> 'my partner and I
      have a dog' is valid, but 'my husband and I have a dog' ->
      'my partner and I have a cat' is not.
  Example: 'my husband and I' -> 'I' is also valid as it only
      removes information.

--User Prompt--
Below I give you a conversation between a user and an AI chatbot
    as well as personal inferences made about the user that can be
     made from the conversation. Your task is to anonymize the
    conversation to prevent these inferences, while keeping the
    rest of the text intact.

Conversation:
{document}

In a first step briefly explain the changes you will make. In a
    new line write a single # and then return the anonymized text.
     Only change relevant parts and keep everything else as is.
```

```
   Make sure you keep the original meaning, i.e. do not invent
   new information instead generalize information.

Inference that can be made from the user (do not include the below
    in the sanitization):
{inference}
```

### C.3.4   SANITIZATION - MEDQA

```
--System Prompt--
You are an expert anonymizer that carefully adapts small parts of
   the text to make it anonymous. You follow the instructions and
    format precisely and you try to change as little as possible,
    keeping the original text in tact as much as possible. Only
   generalize information and do not invent new information.

   Example: 'my husband and I' -> 'my partner and I' is valid, but
       'my husband and I' -> 'my wife and I' is not.
   Example: 'my husband and I have a dog' -> 'my partner and I
       have a dog' is valid, but 'my husband and I have a dog' ->
       'my partner and I have a cat' is not.
   Example: 'my husband and I' -> 'I' is also valid as it only
       removes information.

--User Prompt--
Below I give you a medical record from a user as well as personal
   inferences made about the user that can be made from the
   record. Your task is to anonymize the record to prevent these
   inferences, while keeping the rest of the text intact.

Record:
{document}

In a first step briefly explain the changes you will make. In a
   new line write a single # and then return the anonymized text.
    Only change relevant parts and keep everything else as is.
   Make sure you keep the original meaning, i.e. do not invent
   new information instead generalize information.

Inference that can be made from the user (do not include the below
    in the sanitization):
{inference}
```

### C.4   PRIVACY METRIC

```
**Task**:
For each of the following claims, rate how well the provided
   passage supports it on a scale of 1 to 3, where:
- **1** means "Same Information,"
- **2** means "Different but Similar Information," and
- **3** means "unsupported."

Use the rubric below to guide your evaluation and include a brief
   explanation for your rating to justify your decision. Focus on
```

```
      the content and facts being discussed, ignoring wording
      differences. Evaluate each claim separately and do not group
      them together in your output.

**Rubric**:
1. *Same Information*: The claim exactly matches the corresponding
      information in the passage. The claim could be a direct quote
      from the passage or a paraphrase that captures the same
      information.
2. *Different but Similar Information*: The claim is not identical
      to the passage but captures the same or closely related
      information. For example, the claim could be an abstraction of
      the corresponding information in the passage, or it could
      have minor differences that do not significantly change its
      overall meaning.
3. *Unsupported*: The claim is not supported by the passage. It
      could either be not discussed at all in the passage or
      inconsistent with the passage.

{icl_samples}

**Input**:
{atomized_claims}

Original text: {context}
**Output**:
```

## C.5   QUALITY OF GENERATION

### C.5.1   QUALITY OF GENERATION - MedQA

```
You are provided with a concise medical case summary. Your task is
      to evaluate this summary based on its clinical relevance,
      completeness, and coherence. Use the following 5-point scale:
1 (Very Poor): The summary is severely lacking in critical
      information, contains major contradictions, or is so
      incomplete that it fails to provide a meaningful picture of
      the patient's condition.
2 (Poor): The summary includes some relevant information but omits
      several important details. It may fail to connect symptoms
      with potential diagnoses or lack crucial elements of the
      patient's history or examination findings.
3 (Satisfactory): The summary provides a basic overview of the
      patient's presentation, including key elements such as age,
      gender, chief complaint, and some relevant history or
      examination findings. However, it may lack depth or miss some
      important details.
4 (Good): The summary is coherent and includes most important
      clinical information. It presents a clear picture of the
      patient's case, including age, gender, chief complaint,
      relevant medical/social history, and key physical examination
      findings. Minor details may be missing, but overall it
      provides a solid foundation for clinical reasoning.
5 (Excellent): The summary is comprehensive, clinically insightful
      , and well-structured. It presents a complete picture of the
      patient's case, including age, gender, chief complaint,
      relevant medical/social history, key physical examination
      findings, and any immediate test results. The summary
```

```
      effectively highlights the most clinically relevant
      information and provides a strong basis for forming a
      differential diagnosis.
Evaluation Instructions:

Read the case summary carefully.
Assess the summary based on the above criteria, focusing on its
    clinical relevance, completeness, and coherence.
Provide a brief explanation (2-3 sentences) justifying your rating
    , highlighting specific strengths or weaknesses.
Assign a rating from 1 to 5.

Your response should follow this format:
Explanation: [Your 2-3 sentence justification]
Rating: [Your rating (1-5)]
Remember, you are evaluating the quality and completeness of the
    case summary itself, not making a diagnosis or judging the
    medical decisions. Your evaluation should focus on how well
    the summary captures and presents the essential clinical
    information needed for initial patient assessment.

Medical Profile: {Document}
```

### C.5.2  QUALITY OF GENERATION - WILDCHAT

```
Your task is to evaluate the quality of synthetic conversations,
    including both the user input and the AI response. Each entry
    represents a complete exchange. Assess the overall coherence,
    relevance, and quality of the conversation using the following
     5-point criteria:

Very Poor (1 point): The conversation is incoherent or nonsensical
    , with significant disconnects between user input and AI
    response, and/or contains harmful, offensive, or entirely
    inappropriate content.
Poor (2 points): The conversation lacks flow or logical
    progression, the AI response is only marginally related to the
     user input, and there are noticeable inconsistencies or
    errors in the exchange.
Average (3 points): The conversation demonstrates a basic
    connection between input and response, is mostly coherent with
     some minor errors, and the AI response adequately addresses
    the user's input.
Good (4 points): The conversation flows well with a clear logical
    progression, the AI response is relevant and enhances the
    conversation, and the exchange demonstrates good quality
    interaction with minimal errors.
Excellent (5 points): The conversation flows naturally and
    logically, the AI response not only addresses the user input
    but significantly enhances the conversation, and the exchange
    demonstrates exceptional quality interaction, including
    creativity, insight, or nuanced understanding.

Evaluation Instructions:

Carefully read the entire conversation exchange.
```

```
Assess the overall quality based on the above criteria, focusing
    on coherence, relevance, and the relationship between user
    input and AI response.
Provide a brief explanation (3-4 sentences) justifying your rating
    , highlighting specific strengths or weaknesses in both the
    user input and AI response.
Assign a rating from 1 to 5.

Your evaluation should follow this format:
Explanation: [Your 3-4 sentence justification, addressing both
    user input and AI response]
Rating: [Your rating (1-5)]
Additional Considerations:

Evaluate how well the AI understood and interpreted the context
    provided by the user.
Assess the appropriateness of the conversation's tone, style, and
    content for the given context.
Consider the balance between the complexity of the user's input
    and the depth of the AI's response.
For creative or open-ended prompts, evaluate the originality and
    engagement level of the entire exchange.
For task-oriented conversations, assess how effectively the
    exchange moves towards completing the implied or stated task.

Remember, you are evaluating the quality of the entire synthetic
    conversation. Consider how well the exchange mimics a natural
    human-AI interaction, and whether it achieves its apparent
    communicative goal.

Conversation: {Document}
```

## C.6   UTILITY

### C.6.1   UTILITY - MEDQA

```
Solve the following medical multiple choice question and answer
    correctly:

Question:
{question}
Options:
A. {options['A']}
B. {options['B']}
C. {options['C']}
D. {options['D']}

Answer:
```

### C.6.2   UTILITY - WILDCHAT

```
Read the following conversation between a user and an AI chatbot.
    Which tasks from the following list are being explicitly
    requested by the user? Return only the most likely task name.

Tasks:
- summarization
```

```
- model jailbreaking (e.g. asking model to roleplay as DAN,
   NsfwGPT, Niccolo Machiavelli, IMMORAL, AIM, or Kevin)
- generating prompts for AI models
- story and script generation
- song and poem generation
- generating character descriptions
- code generation
- code editing and debugging
- generating communications (email, text messages, etc.)
- generating non-fictional documents (resumes, essays, etc.)
- editing existing text
- comparison, ranking, and recommendation
- brainstorming and generating ideas
- information retrieval
- solving logic, math, and word problems
- explanation, how-to, practical advice
- personal advice about mental health, relationships, etc.
- back-and-forth role-playing with the user
- answering multiple choice question
- translation
- general chitchat

Conversation:
{context}

Answer:
```

