# OpenReview forum: "A False Sense of Privacy: Evaluating Textual Data Sanitization Beyond Surface-level Privacy Leakage"
_NeurIPS.cc/2024/Workshop/SafeGenAi — SafeGenAi Poster_

### Official Review · Reviewer_CPsF · 2024-10-08
**Good work in privacy privacy assessment for LLMs**

**Rating:** 8
**Confidence:** 4

**Review:**

This work has high quality, clear expression, and strong practical significance, which attempts to evaluate privacy purification methods.

pros:
1. Adequate experiments and interesting key findings
2. The proposed semantic-level privacy is interesting and novel.
3. The introduction of human assessments to enhance credibility
4. The effectiveness of typical protection methods is tested and analyzed

cons:
1. Inadequate privacy-preserving technology categories and the breadth of the dataset limit the generalizability of the results.
2. Why  use this assessment not clarified? Does this kind of assessment guarantee an accurate assessment?
3. More importantly, “appropriate privacy metrics” need to be suggested for real publishing. According to the principle of "Protection as needed", smaller privacy indicators are not better
4. Some generative data privacy efforts can be added: Security and privacy on generative data in aigc: A survey;On protecting the data privacy of large language models (llms): A survey

---

### Official Review · Reviewer_5QPf · 2024-10-09
**Paper challenges the data-sanitization techniques and shows most method still suffer from significant information leakage**

**Rating:** 6
**Confidence:** 3

**Review:**

In this paper, the authors thoroughly evaluate the privacy leakage after applying the data-sanitization methods. In particular, they use auxiliary information to retrieve the most relevant sanitized documents given an auxiliary information about a data-subject and then quantify the privacy information gain from the retrieved records.

**Strengths**
- The findings in the paper challenge mainstream privacy protection methods and open a discussion for a thorough assessment in the future.
- The proposed attack method leverages semantic similarity instead of relying on lexical matching to understand the information gain obtained by retrieving the most relevant sanitized documents.
- The paper conducts experiments on two datasets against four SOTA anonymizers, including the LLM-based anonymizers of Staab (2024).

**Weakness**
- The attack assumes some auxiliary information about the subject.

Overall, the paper contributes to the community by raising awareness of the privacy issues associated with data-sanitization methods, and will be an interesting addition to the workshop.

---

### Official Review · Reviewer_au8U · 2024-10-09
**A False Sense of Privacy: Evaluating Textual Data Sanitization Beyond Surface-level Privacy Leakage**

**Rating:** 5
**Confidence:** 3

**Review:**

In this paper the authors have developed a framework to evaluate the effectiveness of data sanitization methods. Their framework entails a linking attack and a semantic similarity metric to evaluate the privacy protection ability of the sanitizer.

Pro's of the paper :
1. The authors have conducted analysis on a wide range of data sanitization approaches.
2. The framework goes beyond the lexical overlap and includes semantic similarity which provides stricter guarantees on privacy of datasets.

A few questions for the authors :

1. In Table 2, why for different \epsilon values the sematic similarity numbers are similar but the drop in the task utility for WildChat dataset goes down from 0.88 to 0.7?
2. The results show in Table 3 that there is inconsistent correlation between the auxiliary information known to the attacker It will be untersting to understand the importance of the type of auxiliary information and an approach to identify what auxiliary information may be the most important - maybe a few anecdotes might help to understand what type of auxiliary information causes the biggest privacy leak.
3. In the experiments the authors have assumed that the auxiliary information comes as claims from the original dataset. It would be interesting to see how their framework does if this assumption is true as this is not reflective of the real world scenario. A way to experiment with this would be to maybe perturb the auxiliary information and understand if their framework can still capture that privacy leakage.